# The Association of Dining Companionship with Energy and Nutrient Intake Among Community-Dwelling Japanese Older Adults

**DOI:** 10.3390/nu17010037

**Published:** 2024-12-26

**Authors:** Yuki Minagawa-Watanabe, Shigekazu Ukawa, Tomoe Fukumura, Satoe Okabayashi, Masahiko Ando, Kenji Wakai, Kazuyo Tsushita, Akiko Tamakoshi

**Affiliations:** 1Graduate School of Human Life and Ecology, Osaka Metropolitan University, 1-1-138 Sugimoto, Sumiyoshi-ku, Osaka 558-8585, Japan; sj24587w@st.omu.ac.jp (Y.M.-W.); fukumura@omu.ac.jp (T.F.); 2Agency for Health, Safety and Environment, Kyoto University, Kyoto 606-8501, Japan; okabayashi.satoe.8c@kyoto-u.ac.jp; 3Center for Advanced Medicine and Clinical Research, Nagoya University Hospital, Nagoya 466-8560, Japan; mando@med.nagoya-u.ac.jp; 4Department of Preventive Medicine, Graduate School of Medicine, Nagoya University, Nagoya 466-8550, Japan; wakai@med.nagoya-u.ac.jp; 5Graduate School of Nutrition Sciences, Kagawa Nutrition University, Sakado 350-0288, Japan; tsushita.kazuyo@eiyo.ac.jp; 6Department of Public Health, Faculty of Medicine, Hokkaido University, Sapporo 060-8638, Japan; tamaa@med.hokudai.ac.jp

**Keywords:** meal, companion, social facilitation, dietary intake, aged, community dwelling

## Abstract

Background: Community-dwelling older adults are at risk of malnutrition due to age-related declines in energy and nutrient intake. While the positive effect of dining companions on energy and nutrient intake has been suggested, evidence remains inconclusive. This study investigated the association between the number of dining companions and energy and nutrient intake, as well as the contribution of specific food groups to higher intake in the presence of dining companions. Methods: This cross-sectional study included 2865 community-dwelling older adults. The number of dining companions was assessed through self-administered questionnaires and categorized into three groups (none, 1, ≥2). Dietary intake was evaluated using a validated food frequency questionnaire, and multivariable regression analyses were conducted to control for potential confounders. Results: Participants dining with two or more companions consumed significantly more energy (β 143.85; 95% CI: 30.05, 257.65; *p* for trend = 0.01), protein (β 6.32; 95% CI: 1.05, 11.59), fat (β 6.78; 95% CI: 2.44, 11.12; *p* for trend = 0.002), and carbohydrates (β 17.43; 95% CI: 1.48, 33.37; *p* for trend = 0.06) compared to those dining alone. They also consumed higher amounts of rice, fats and oils, meat, other vegetables, fruits, and mushrooms. Conclusions: Dining with two or more companions is associated with greater energy and nutrient intake, particularly from energy- and nutrient-dense foods. Encouraging shared meals could serve as a potential approach to support dietary quality and address risks of malnutrition in older adults.

## 1. Introduction

The amounts of energy and nutrients that community-dwelling older adults consume decrease with age and tend to be inadequate [1,2,3]. This may predispose older adults to weight loss, undernutrition, and protein energy malnutrition. Although energy requirements decrease with age as a result of decreased energy expenditure and physical activity [4], the fall in the amount of energy intake is often greater than the decrease in energy requirements. Nutrient requirements do not necessarily decrease with age and even tend to remain the same or increase [5]. For example, protein, which is one of the macro-nutrients, is recommended to be increased for older adults more than for other age groups to alleviate age-related conditions such as skeletal muscle loss [6,7]. Multiple factors that contribute to the decreased amount of energy and nutrient intake in older adults are known to include sensory changes, anorexia, diseases, medication use, and psychological factors such as depression [1,3,8].

Japan is facing an aging society, with family structures transitioning from extended families to nuclear families and, more recently, to single-person and couple households. Among older adults in particular, single-person and couple households accounted for 63.9% of all individuals aged 65 and over in 2022, a significant increase from 31.3% in 1986 [9]. As a result of changes in family structure, older adults have fewer opportunities to eat with two or more companions. Eating alone has been reported to be linked to unhealthy dietary behaviors [10]. Studies focusing on eating with companions have shown a positive correlation between the number of eating companions and the intake of energy and nutrients among younger individuals [11,12,13,14,15,16,17,18,19,20,21,22]. However, research focusing on older adults is limited [23,24,25] and has not yet established a distinct association between the number of eating companions and the intake of energy and nutrients because prior studies have not controlled for confounders associated with energy intake.

Although social facilitation, the influence of others on individual behavior [26], has been presented as a plausible reason why people consume more energy when dining with others than when eating alone, which food groups contribute to the increase in the amount of energy and nutrient intake are still not revealed. Because specific food groups, such as vegetables and fruits, were reported to have an inverse association with malnutrition and unhealthy aging [27,28], focusing on intake from each food group is also needed. Therefore, the aim of this study is to confirm the association between the number of dining companions and energy and nutrient intake, particularly macro-nutrients (protein, carbohydrate, and fat), while controlling for confounders associated with energy intake. In addition, we seek to identify the specific food groups that contribute to this association in the context of dining with more dining companions, focusing on Japanese older adults.

## 2. Methods

### 2.1. Study Population

Participants in this study were drawn from the New Integrated Suburban Seniority Investigation Project, an age-specific research project involving residents of Nisshin City, located in central Japan [29]. Community-dwelling older adults aged 64–65 years were invited by letter to participate in a free comprehensive medical health checkup and to complete a detailed questionnaire survey annually from 1996 to 2005. Among 3073 potential participants, those with missing data on dietary examinations (*n* = 97) or the number of dining companions (*n* = 73) were excluded. Additionally, we excluded those whose estimated total energy intake fell outside three standard deviations above or below the sex-specific mean (*n* = 38). As a result, 2865 individuals (1433 men and 1432 women) remained for the final analyses (Figure 1). This study was conducted in accordance with the guidelines laid down in the Declaration of Helsinki, and all procedures involving the study participants were approved by the Ethics Committees of Nagoya University Graduate School of Medicine, the National Center for Geriatrics and Gerontology of Japan, Aichi Medical University School of Medicine, Hokkaido University Graduate School of Medicine, and Osaka City University Graduate School of Human Life Science. Informed consent was obtained from all participants for their involvement in the study, including any follow-up procedures. An opt-out approach was applied from 1996 to 2001. Health checkup examinees who declined to participate in the study were recorded and excluded from the survey. From 2002 to 2005, participants provided written consent through an opt-in approach.

### 2.2. Data Collection

Information was collected via self-reported questionnaires and objective assessments during the health check-ups.

Information on the number of dining companions was collected through self-administered questionnaires using the following question: “In the past year, excluding yourself, how many people usually dined with you on weekdays?” We then categorized the number of dining companions into three groups (none, 1, ≥2) and set none as a reference category based on previously identified associations with adverse health-related outcomes [10,30,31,32].

A food frequency questionnaire (FFQ) was employed to assess the intake of 16 food groups including rice, bread, noodles, sugars and sweeteners, confectioneries, fats and oils, nuts and seeds, pulses, fish and shellfish, meats, eggs, milk and dairy products, green-yellow vegetables, other vegetables, fruits, and mushrooms. The FFQ, which was designed to reflect modern Japanese foods and dishes, covered 97 food items. Energy, protein, fat, and carbohydrate intake was also evaluated. The frequency of consumption for each food item was recorded using an incremental scale (<once/month, once/month, 2–3 times/month, once/week, 2–4 times/week, 5–6 times/week, once/day, 2–3 times/day, ≥4 times/day). Except for rice, portion sizes were fixed across all food items. The validity of the FFQ in measuring food group and nutrient intake has been confirmed in previous studies [33,34].

### 2.3. Statistical Analysis

The characteristics of the participants were compared across each category of dining companions. Categorical variables were analyzed using the chi-squared test. To explore the relationship between the number of dining companions and the intake of 16 food groups, multiple regression models were employed. Multivariable regression coefficients (β) and 95% confidence intervals (CIs) were estimated, with individuals having no dining companions as the reference group. The regression models were adjusted for potential confounders. The following covariates were included step-by-step as confounders in the regression models: enrollment year (1996–1999, 2000–2002, 2003–2005), sex (men or women) (model 1); educational attainment (lower than high school, high school or above or other/missing), cohabitation status (living alone, living with others, or missing), smoking habits (current, past, or never), alcohol consumption (current, never/past, or missing), walking status (<1, ≥1 h/day, or missing), depressive symptoms (presence, absence, or missing), instrumental activities of daily living (IADL: deteriorated, not deteriorated, or missing), history of hypertension (yes or no/missing), diabetes mellitus (yes or no/missing), cardiovascular diseases (yes or no/missing), cancer (yes or no/missing), hyperlipidemia (yes or no/missing) (model 2); body mass index (BMI), categorized as <18.5, 18.5–22.9, ≥23 kg/m^2^ [35] (model 3). BMI was calculated as weight in kilograms divided by height in meters squared. Depressive symptoms were assessed using the short-form Geriatric Depression Scale (GDS), a tool validated for older Japanese populations [36]. A GDS score of less than 6 points was classified as indicative of depressive symptoms [37]. The Instrumental Activity of Daily Living index measures 5 functions: public transportation, shopping, meal preparation, management of financial matters, and handling of money in the bank. Participants unable to perform at least one of these functions were considered to have significant impairment [38]. Hypertension was defined as systolic blood pressure ≥ 140 mmHg, diastolic blood pressure ≥ 90 mmHg, self-reported hypertension, and/or antihypertensive medication. Diabetes mellitus was defined by fasting blood glucose ≥ 126 mg/dL, hemoglobin A1c (HbA1c) ≥ 6.5%, self-reported diabetes mellitus, and/or antidiabetic medication. HbA1c values were converted to National Glycohemoglobin Standardization Program (NGSP) equivalents using the formula: NGSP (%) = (1.02 × JDS (%)) + 0.25%, as the HbA1c levels were originally measured according to Japanese Diabetes Society criteria [39]. Cerebrovascular disease was classified as self-reported stroke, cerebral hemorrhage, and subarachnoid hemorrhage. Cancer was defined by self-reported diagnosis, and hyperlipidemia was characterized by a total cholesterol level ≥ 220 mg/dL, self-reported hyperlipidemia, and/or hyperlipidemia medication. Then, the analyses were repeated for intake of energy, protein, fat, and carbohydrate. A trend test was performed using the original number of dining companions as a continuous variable in the multivariable regression model. All statistical tests were two-sided, and an alpha level of 0.05 was considered statistically significant. All statistical analyses were conducted using EZR 1.61 [40].

## 3. Results

The participants consumed an average of 1882 ± 570 kcal/day. The mean intake of protein, fat, and carbohydrate intakes was protein 72.2 ± 26.7 g/day, fat 52.8 ± 21.9 g/day, and carbohydrate 255.1 ± 79.8 g/day, respectively. The participants had mean amounts of food groups as follows: rice 40.07 ± 19.62 g/day, bread 22.73 ± 21.74 g/day, noodles 97.41 ± 86.86 g/day, sugars and sweeteners 4.12 ± 2.30 g/day, confectioneries 16.37 ± 16.81 g/day, fats and oils 10.76 ± 6.24 g/day, nuts and seeds 2.58 ± 3.42 g/day, pulses 79.00 ± 51.41 g/day, fish and shellfish 77.77 ± 55.14 g/day, meats 52.02 ± 38.47 g/day, eggs 33.21 ± 26.59 g/day, milk and dairy products 182.35 ± 144.94 g/day, green-yellow vegetables 118.63 ± 95.90 g/day, other vegetables 123.96 ± 67.45 g/day, fruits 200.63 ± 139.37 g/day, and mushrooms 11.96 ± 10.12 g/day.

The characteristics of the study participants according to the number of dining companions are shown in Table 1. Males were more represented in the one dining companion group. Participants with lower education levels were more commonly found among those with two or more dining companions. Most participants living alone were in the no dining companions group, whereas nearly all participants with dining companions lived with others. Current smokers were most numerous in the no dining companions group. Current alcohol consumption was highest among those with one dining companion. Walking time was shortest in the one dining companion group. Depressive symptoms were more frequently observed in the no dining companions group compared to the other groups. IADL deterioration was more typical in the group with two or more dining companions. No significant differences were observed in medical history or BMI across the groups.

The associations of the number of dining companions with food group intake are shown in Table 2. Significant positive associations were found between the number of dining companions and the intake of various food groups. Participants who dined with two or more companions had a higher intake of rice (*p* for trend = 0.003), fats and oils (β 1.99; 95% CI: 0.74, 3.23; *p* for trend < 0.001), meat (β 7.90; 95% CI: 0.25, 15.55; *p* for trend < 0.001), other vegetables (β 13.84; 95% CI: 0.56, 27.12; *p* for trend = 0.31), fruits (β 30.86; 95% CI: 5.09, 56.62; *p* for trend = 0.01), and mushrooms (β 2.48; 95% CI: 0.53, 4.44; *p* for trend = 0.08) compared to those who dined alone, after adjusting for potential confounders. Milk and dairy products (β 37.45; 95% CI: 10.31, 64.60) and green-yellow vegetables (β 24.77; 95% CI: 6.76, 42.79) were significantly more consumed by participants dining with one companion, but these associations were not significant for those dining with two or more companions.

Table 3 shows the association between the number of dining companions and the intake of energy and nutrients. Participants who have two or more dining companions consume more energy (β 143.85; 95% CI: 30.05, 257.65; *p* for trend = 0.01), protein (β 6.32; 95% CI: 1.05, 11.59), fat (β 6.78; 95% CI: 2.44, 11.12; *p* for trend = 0.002), and carbohydrates (β 17.43; 95% CI: 1.48, 33.37; *p* for trend = 0.06) compared to those who dined alone, after adjusting for potential confounders.

## 4. Discussion

The present study found a significant positive association between the number of dining companions and the intake of energy and nutrients among community-dwelling older adults. Participants dining with two or more companions consumed higher amounts of energy, protein, fat, and carbohydrates than those dining alone. Furthermore, those dining with more companions had higher intake of rice, fats and oils, meat, fruits, other vegetables and mushrooms, which likely contributed to the observed increase in nutrient intake.

Our results are consistent with previous studies that suggested a positive correlation between the number of eating companions and the energy intake, based on data collected from 7-day food diaries for each meal [11,25]. A study that followed 153 adults aged 18–67 years in America found that the amounts of energy, protein, fat, and carbohydrate intake increase as a function of the number of eating companions, in a fashion best described by a power function [25]. Another study that followed 56 French participants with Type I diabetes and 28 healthy controls (52.5 ± 1.9 years of age, range 19–77) found that both groups had a positive correlation between the number of people present at the meal and amounts of energy intake [24]. In an observational study of 348 men and 414 women in America, participants were divided into four groups: 20–34 years old (*n* = 325), 35–49 years old (*n* = 292), 50–64 years old (*n* = 99), and over 65 years (*n* = 46). All four groups had a positive relationship between the number of people present at the meal and amount of energy intake [23]. While these studies share similarities with ours in investigating the associations between the number of dining companions and the amount of intake, our study offers a distinct contribution by controlling the potential confounders that are likely to affect energy intake for community-dwelling older adults.

A potential mechanism by which having more dining companions leads to higher energy intake may be social facilitation, defined as the phenomenon where individual behavior is influenced by the presence of other people engaged in the same behavior [26]. Eating behavior is one of the clearest demonstrations of the social facilitation effect. People eat more in groups than when alone, which is called social facilitation of eating. There are two primary hypotheses in the literature regarding the mechanism of social facilitation of eating. The arousal hypothesis states that in larger groups activation or arousal is greater, resulting in a faster eating rate and greater consumption [41,42]. The time extension hypothesis suggests that larger group size increases social interaction, extends meal duration, and increases the length of time an individual is in the presence of food, thereby increasing intake [16,41]. In addition, another reason why energy intake may be higher when increasing the number of dining companions is the greater chance to get social support from other people, including family and friends, neighbors, and caregivers. They may provide encouragement and support to eat more, even when the participants are in difficult situations to eat, such as illness and poor appetite, or feeling depressed.

Our study found that participants dining with two or more companions consumed larger amounts of fats and oils, which are high-energy-density foods [43,44], compared to participants dining with fewer companions. They also ate more rice, meat, fruits, other vegetables and mushrooms. Although the reasons for greater consumption of specific food groups, such as rice, fats and oils, meat, fruits, other vegetables and mushrooms, among participants dining with two or more companions remain unclear, the diversity of those dining together, especially when the group includes individuals from different age groups, may also influence the types of food consumed. For instance, the presence of younger individuals during meals might encourage older individuals to consume more fats, as well as vegetables and fruits, which are considered healthy foods. The fat intake between younger and older individuals in the same household showed a weak to moderate positive correlation [45,46,47]. Another study has shown that younger individuals encourage older individuals in the household to adopt healthy habits [48,49,50]. However, we did not obtain information on family composition or with whom participants usually dined, so future studies are needed to clarify this point.

The present study has some notable strengths. We were able to eliminate the negative effect of aging on the association between the number of dining companions and the amount of intake by using data from age-specific community-dwelling older adults because previous studies have shown that aging is associated with a lower number of dining companions [23] and lower intake of energy and nutrients [1]. Furthermore, to enhance the robustness of our findings, we employed multivariable regression analyses to control for confounders known to be related to energy intake. These confounders included educational level [51], living arrangement [52], smoking status [53], alcohol consumption status [54], physical activity [55], depressive symptoms [56], IADL [57], medical histories [1], and BMI [58], which were not accounted for in previous studies [23,24,25]. However, certain limitations to the study warrant consideration. First, due to the cross-sectional nature, we could not identify temporal relationships between the number of dining companions and the amounts of intake. A possibility of reverse causality exists, where individuals who inherently consume higher energy levels may choose to dine with more companions. Social interactions can contribute to the enjoyment of meals, which limit the ability to draw causal conclusions. To clarify such temporal relationships, conducting a cohort study would be required. Second, although we collected information on the amounts of dietary intake for breakfast, lunch, and dinner, the number of eating companions was obtained only for dinner. This discrepancy should be taken into account when interpreting the results. A previous study of community-dwelling men and women in the Netherlands showed that the mean number of eating companions at breakfast, lunch, and dinner was 1.1 ± 1.2, 2.5 ± 2.5, and 3.6 ± 2.3, respectively [16]. Therefore, an overestimation of the number of eating companions may have occurred in this study, which may have led to an underestimation of our results. Third, although the FFQ has been validated for dietary intake amounts through a 4-day dietary record [34], the use of self-reported dietary questionnaires may still have led to inaccurate estimations of dietary intake. Fourth, the use of self-reported lifestyle questionnaires may introduce some information bias.

While this study focused on the association between the number of dining companions and the amount of dietary intake, dietary quality, including dietary diversity, has also been reported to be inversely related to the risk of cancer and all-cause mortality among older adults [59,60]. The positive association between the number of dining companions and fat and oil intake remained even after further adjusting for energy intake in the regression model; however, future studies are needed to clarify the association between the number of dining companions and dietary quality. Furthermore, while this study hypothesized social facilitation as a potential mechanism, future research is needed to focus on the role of social interactions during meals.

## 5. Conclusions

This study showed dining with two or more companions was associated with greater energy intake and consumption of protein, fat, and carbohydrates from rice, fats and oils, meat, fruit, other vegetables and mushrooms. This highlights the role of social facilitation in eating behavior, suggesting that dining with companions may serve as a practical intervention for improving dietary quality and preventing malnutrition in older adults.

## Figures and Tables

**Figure 1 nutrients-17-00037-f001:**
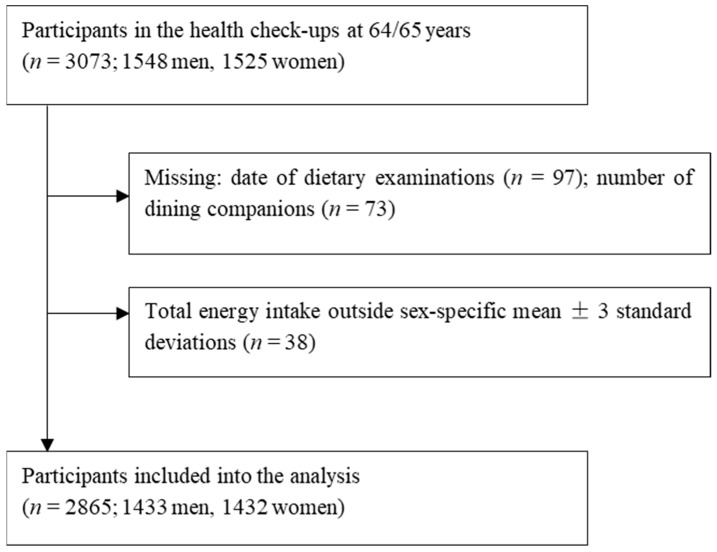
Participant flow of the present study.

**Table 1 nutrients-17-00037-t001:** Characteristics of study participants according to number of dining companions (*n* = 2865).

Variables	Number of Dining Companions			
	None (*n* = 194)	1 (*n* = 1872)	≥2 (*n* = 799)	*p*-Value
Sex				
Male	78 (40.2)	986 (52.7)	369 (46.2)	<0.001
Female	116 (59.8)	886 (47.3)	430 (53.8)	
Highest level of education				
Junior high school or below	59 (30.4)	518 (27.7)	317 (39.7)	<0.001
High school or above	135 (69.6)	1350 (72.1)	477 (59.7)	
Living arrangement				
Living alone	96 (49.5)	8 (0.4)	4 (0.5)	<0.001
Living with others	92 (47.4)	1827 (97.6)	773 (96.7)	
Smoking status				
Current	38 (19.6)	310 (16.6)	156 (19.5)	0.003
Past	44 (22.7)	545 (29.1)	178 (22.3)	
Never	112 (57.7)	1017 (54.3)	465 (58.2)	
Alcohol consumption status				
Current	73 (37.6)	869 (46.4)	323 (40.4)	0.02
Never/Past	121 (62.4)	1002 (53.5)	476 (59.6)	
Walking status (h/day)				
<1	79 (40.7)	825 (44.1)	316 (39.5)	0.02
≥1	115 (59.3)	1040 (55.6)	474 (59.3)	
Depressive symptoms				
Presence	55 (28.4)	369 (19.7)	194 (24.3)	0.01
Absence	139 (71.6)	1495 (79.9)	602 (75.3)	
IADL				
Deteriorated	7 (3.6)	219 (11.7)	104 (13.0)	0.01
Not deteriorated	187 (96.4)	1.648 (88.0)	693 (86.7)	
Medical history of				
Hypertension				
Presence	101 (52.1)	863 (46.1)	351 (43.9)	0.12
Diabetes mellitus				
Presence	17 (8.8)	207 (11.1)	73 (9.1)	0.25
Cardio-cerebrovascular diseases				
Presence	5 (2.6)	60 (3.2)	25 (3.1)	0.89
Cancer				
Presence	5 (2.6)	75 (4.0)	30 (3.8)	0.618
Hyperlipidemia				
Presence	27 (13.9)	250 (13.4)	109 (13.6)	0.96
BMI (kg/m^2^)				
<18.5	6 (3.1)	94 (5.0)	30 (3.8)	0.37
18.5–22.9	98 (50.5)	882 (47.1)	369 (46.2)	
≥23.0	90 (46.4)	896 (47.9)	400 (50.1)	

IADL, instrumental activity of daily living; BMI, body mass index. Values are expressed as number (percentage). *p*-values were calculated with chi-square tests. The proportion of each variable does not always add up to 100 owing to missing data.

**Table 2 nutrients-17-00037-t002:** Associations of number of dining companions with food group intake (*n* = 2865).

	Number of Dining Companions			
	None (*n* = 194)	1 (*n* = 1872)	≥2 (*n* = 799)	*p* for Trend
Rice (g)				
β (95% CI) ^1^	Ref.	0.67 (−2.14, 3.50)	3.90 (0.91, 6.90) *	<0.001
β (95% CI) ^2^	Ref.	−0.51 (−4.19, 3.15)	2.23 (−1.57, 6.04)	0.004
β (95% CI) ^3^	Ref.	−0.58 (−4.26, 3.08)	2.23 (−1.57, 6.04)	0.003
Bread (g)				
β (95% CI) ^1^	Ref.	0.88 (−2.33, 4.10)	−0.70 (−4.12, 2.71)	0.15
β (95% CI) ^2^	Ref.	0.26 (−3.92, 4.46)	−0.88 (−5.24, 3.46)	0.32
β (95% CI) ^3^	Ref.	0.31 (−3.88, 4.51)	−0.81 (−5.16, 3.53)	0.37
Noodles (g)				
β (95% CI) ^1^	Ref.	9.07 (−3.80, 21.94)	6.63 (−7.01, 20.28)	0.99
β (95% CI) ^2^	Ref.	7.88 (−8.92, 24.70)	5.78 (−11.65, 23.22)	0.90
β (95% CI) ^3^	Ref.	8.08 (−8.73, 24.90)	6.00 (−11.43, 23.44)	0.94
Sugars and sweeteners (g)				
β (95% CI) ^1^	Ref.	0.29 (−0.04, 0.63)	0.15 (−0.20, 0.51)	0.75
β (95% CI) ^2^	Ref.	0.33 (−0.11, 0.77)	0.22 (−0.23, 0.68)	0.94
β (95% CI) ^3^	Ref.	0.32 (−0.11, 0.77)	0.23 (−0.22, 0.68)	0.96
Confectioneries (g)				
β (95% CI) ^1^	Ref.	1.37 (−1.04, 3.80)	0.51 (−2.05, 3.08)	0.06
β (95% CI) ^2^	Ref.	0.73 (−2.40, 3.87)	0.27 (−2.98, 3.53)	0.13
β (95% CI) ^3^	Ref.	0.72 (−2.41, 3.87)	0.27 (−2.98, 3.53)	0.13
Fats and oils (g)				
β (95% CI) ^1^	Ref.	0.78 (−0.13, 1.70)	2.13 (1.15, 3.11) *	<0.001
β (95% CI) ^2^	Ref.	0.59 (−0.60, 1.79)	1.99 (0.74, 3.24) *	<0.001
β (95% CI) ^3^	Ref.	0.59 (−0.61, 1.79)	1.99 (0.74, 3.23) *	<0.001
Nuts and seeds (g)				
β (95% CI) ^1^	Ref.	0.30 (−0.19, 0.81)	0.17 (−0.35, 0.71)	0.62
β (95% CI) ^2^	Ref.	0.55 (−0.10, 1.21)	0.45 (−0.23, 1.13)	0.84
β (95% CI) ^3^	Ref.	0.55 (−0.10, 1.21)	0.45 (−0.23, 1.13)	0.84
Pulses (g)				
β (95% CI) ^1^	Ref.	9.17 (1.61, 16.73) *	10.63 (2.62, 18.65) *	0.61
β (95% CI) ^2^	Ref.	7.56 (−2.32, 17.45)	8.80 (−1.45, 19.06)	0.9
β (95% CI) ^3^	Ref.	7.46 (−2.43, 17.36)	8.76 (−1.50, 19.02)	0.92
Fish and shellfish (g)				
β (95% CI) ^1^	Ref.	8.08 (−0.03, 16.19)	5.89 (−2.71, 14.49)	0.15
β (95% CI) ^2^	Ref.	8.56 (−2.02, 19.16)	7.14 (−3.84, 18.13)	0.17
β (95% CI) ^3^	Ref.	8.33 (−2.26, 18.93)	7.01 (−3.96, 18.00)	0.18
Meat (g)				
β (95% CI) ^1^	Ref.	3.78 (−1.84, 9.41)	11.13 (5.16, 17.11) *	<0.001
β (95% CI) ^2^	Ref.	0.67 (−6.70, 8.04)	7.80 (0.15, 15.45) *	<0.001
β (95% CI) ^3^	Ref.	0.80 (−6.57, 8.18)	7.90 (0.25, 15.55) *	<0.001
Eggs (g)				
β (95% CI) ^1^	Ref.	4.52 (0.57, 8.46) *	2.93 (−1.24, 7.12)	0.93
β (95% CI) ^2^	Ref.	1.44 (−3.70, 6.58)	0.24 (−5.08, 5.58)	0.73
β (95% CI) ^3^	Ref.	1.49 (−3.65, 6.64)	0.28 (−5.05, 5.62)	0.74
Milk and dairy products (g)				
β (95% CI) ^1^	Ref.	2.89 (−18.22, 24.01)	−19.76 (−42.15, 2.63)	<0.001
β (95% CI) ^2^	Ref.	36.85 (9.71, 63.99) *	19.40 (−8.73, 47.54)	0.05
β (95% CI) ^3^	Ref.	37.45 (10.31, 64.60) *	19.86 (−8.26, 48)	0.06
Green-yellow vegetables (g)				
β (95% CI) ^1^	Ref.	16.32 (2.37, 30.28) *	3.75 (−11.03, 18.55)	0.02
β (95% CI) ^2^	Ref.	24.54 (6.52, 42.56) *	15.67 (−3, 34.36)	0.17
β (95% CI) ^3^	Ref.	24.77 (6.76, 42.79) *	15.97 (−2.7, 34.65)	0.19
Other vegetables (g)				
β (95% CI) ^1^	Ref.	22.61 (12.82, 32.39) *	24.03 (13.65, 34.4) *	0.03
β (95% CI) ^2^	Ref.	12.14 (−0.66, 24.94)	13.84 (0.56, 27.12) *	0.29
β (95% CI) ^3^	Ref.	12.23 (−0.58, 25.04)	13.84 (0.56, 27.12) *	0.31
Fruits (g)				
β (95% CI) ^1^	Ref.	45.26 (25.81, 64.71) *	21.86 (1.24, 42.48) *	0.007
β (95% CI) ^2^	Ref.	49.04 (24.2, 73.88) *	30.60 (4.84, 56.36) *	0.01
β (95% CI) ^3^	Ref.	49.22 (24.37, 74.07) *	30.86 (5.09, 56.62) *	0.01
Mushrooms (g)				
β (95% CI) ^1^	Ref.	2.60 (1.16, 4.05) *	2.81 (1.28, 4.34) *	0.05
β (95% CI) ^2^	Ref.	2.14 (0.26, 4.02) *	2.50 (0.54, 4.45) *	0.09
β (95% CI) ^3^	Ref.	2.11 (0.22, 3.99) *	2.48 (0.53, 4.44) *	0.08

β, multivariable regression coefficients; CI, confidence interval; Ref., Reference. * *p* < 0.05. *p* for trend is calculated across the number of dining companions (continuous). ^1^ adjusted for enroll year, sex. ^2^ adjusted for enroll year, sex, highest level of education, living arrangement, smoking status, alcohol consumption status, walking status, depression symptoms, IADL, medical histories of hypertension, diabetes mellitus, cardio-cerebrovascular diseases, cancer, and hyperlipidemia. ^3^ adjusted for enroll year, sex, highest level of education, living arrangement, smoking status, alcohol consumption status, walking status, depression symptoms, IADL, medical histories of hypertension, diabetes mellitus, cardio-cerebrovascular diseases, cancer, and hyperlipidemia, and BMI.

**Table 3 nutrients-17-00037-t003:** Associations of number of dining companions with nutrient intake (*n* = 2865).

	Number of Dining Companions			
	None (*n* = 194)	1 (*n* = 1872)	≥2 (*n* = 799)	*p* for Trend
Energy (kcal)				
β (95% CI) ^1^	Ref.	116.08 (31.62, 200.54) *	161.14 (71.59, 250.69) *	0.01
β (95% CI) ^2^	Ref.	90.66 (−19.10, 200.42)	142.87 (29.05, 256.70) *	0.02
β (95% CI) ^3^	Ref.	90.25 (−19.51, 200.02)	143.85 (30.05, 257.65) *	0.01
Protein (g)				
β (95% CI) ^1^	Ref.	6.00 (2.10, 9.90) *	6.35 (2.21, 10.48) *	0.38
β (95% CI) ^2^	Ref.	5.52 (0.44, 10.60) *	6.29 (1.02, 11.56) *	0.42
β (95% CI) ^3^	Ref.	5.50 (0.42, 10.59) *	6.32 (1.05, 11.59) *	0.38
Fat (g)				
β (95% CI) ^1^	Ref.	4.18 (0.97, 7.39) *	6.16 (2.75, 9.56) *	0.004
β (95% CI) ^2^	Ref.	4.37 (0.18, 8.55) *	6.74 (2.40, 11.08) *	0.002
β (95% CI) ^3^	Ref.	4.39 (0.21, 8.58) *	6.78 (2.44, 11.12) *	0.002
Carbohydrate (g)				
β (95% CI) ^1^	Ref.	16.30 (4.49, 28.11) *	21.54 (9.02, 34.06) *	0.03
β (95% CI) ^2^	Ref.	11.68 (−3.70, 27.06)	17.26 (1.30, 33.21) *	0.09
β (95% CI) ^3^	Ref.	11.59 (−3.78, 26.97)	17.43 (1.48, 33.37) *	0.06

β, multivariable regression coefficients; CI, confidence interval; Ref., Reference. * *p* < 0.05. *p* for trend is calculated across the number of dining companions (continuous). ^1^ adjusted for enroll year, sex. ^2^ adjusted for enroll year, sex, highest level of education, living arrangement, smoking status, alcohol consumption status, walking status, depression symptoms, IADL, medical histories of hypertension, diabetes mellitus, cardio-cerebrovascular diseases, cancer, and hyperlipidemia. ^3^ adjusted for enroll year, sex, highest level of education, living arrangement, smoking status, alcohol consumption status, walking status, depression symptoms, IADL, medical histories of hypertension, diabetes mellitus, cardio-cerebrovascular diseases, cancer, and hyperlipidemia, and BMI.

## Data Availability

The authors do not have permission to share data.

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
