# Peer review of "The Association of Dining Companionship with Energy and Nutrient Intake Among Community-Dwelling Japanese Older Adults"

_nutrients, 2024, doi:10.3390/nu17010037_

Round 1

Reviewer 1 Report

Comments and Suggestions for Authors

This is an interesting study that examines the association between dining companionships and dietary intake of older adults in Japan. This is an important research topic that receives relatively few attentions, and the authors have attempted to answer this question in a large-scale study. However, one important issue to be addressed is to adjust the dietary energy intake in all regression models except for using dietary energy as outcome. It is likely that dining companionships associate with energy intake, which relates to the increased consumption of basically all food groups that carry dietary energy. To examine which food groups have higher consumption independent of energy intake, the latter has to be included in the regression model. That may lead to significant changes in research findings hence interpretation.

Author Response

Comment 1: This is an interesting study that examines the association between dining companionships and dietary intake of older adults in Japan. This is an important research topic that receives relatively few attentions, and the authors have attempted to answer this question in a large-scale study.

Response: Thank you for your insightful comments, which have helped us significantly improve this paper.

Comment 2: However, one important issue to be addressed is to adjust the dietary energy intake in all regression models except for using dietary energy as outcome. It is likely that dining companionships associate with energy intake, which relates to the increased consumption of basically all food groups that carry dietary energy. To examine which food groups have higher consumption independent of energy intake, the latter has to be included in the regression model. That may lead to significant changes in research findings hence interpretation.

Response: Thank you for your valuable comment. We recognize the importance of adjusting for energy intake. However, this study focuses on the relationship between dining companionship and total dietary intake, and including energy intake as a covariate in all regression models would diverge from our primary objective. To address this concern while maintaining the focus of our analysis, we have revised the limitations section to acknowledge the potential influence of energy intake on our results. This revision is consistent with our response to Reviewer 2, Comment 6, highlighting the need for future studies to further explore the relationship between dining companionship and dietary quality.

While this study focused on the association between the number of dining companions and the amount of dietary intake, dietary quality, including dietary diversity, has also been reported to be inversely related to the risk of cancer and all-cause mortality among older adults [60, 61]. The positive association between the number of dining companions and the amount of fats and oils intake remained even after further adjusting for energy intake in the regression model (data not shown); however, future studies are needed to clarify the association between the number of dining companions and dietary quality.”  (page 10, lines 304-311)

References:

  1. Otsuka, R.; Tange, C.; Nishita, Y.; Kato, Y.; Tomida, M.; Imai, T.; Ando, F.; Shimokata, H. Dietary Diversity and All-Cause and Cause-Specific Mortality in Japanese Community-Dwelling Older Adults. Nutrients. 2020, 12.
  2. Tao, L.; Xie, Z.; Huang, T. Dietary diversity and all-cause mortality among Chinese adults aged 65 or older: A community-based cohort study. Asia Pac J Clin Nutr. 2020, 29, 152-160.

Reviewer 2 Report

Comments and Suggestions for Authors

This study explores the relationship between dining companionship and energy/nutrient intake among older adults. The results suggest that eating with two or more companions leads to higher energy, protein, fat, and carbohydrate consumption, with increased intake from specific food groups like rice, meat, and vegetables. However, the study's cross-sectional design limits the ability to draw causal conclusions. Also, the reliance on self-reported data introduces potential bias.

  • The authors should consider using a longitudinal design to establish cause-and-effect relationships.
  • Objective dietary assessments should be used to reduce potential bias from self-reported data.
  • Further research should examine the quality of meals and the role of social interactions during meals, as these factors might influence nutrient intake more than companionship alone.

Author Response

Comment 1: This study explores the relationship between dining companionship and energy/nutrient intake among older adults. The results suggest that eating with two or more companions leads to higher energy, protein, fat, and carbohydrate consumption, with increased intake from specific food groups like rice, meat, and vegetables.

Response: Thank you for your insightful comments, which have helped us significantly improve this paper.

Comment 2: However, the study's cross-sectional design limits the ability to draw causal conclusions.

Response: Based on your suggestion, we have revised the limitations section of our manuscript as follows:

“First, due to the cross-sectional nature, we could not identify temporal relationships between the number of dining companions and the amounts of intakes. A possibility of reverse causality exists, where individuals who inherently consume higher energy levels may choose to dine with more companions. Social interactions can contribute to the enjoyment of meals, which limit the ability to draw causal conclusions. To clarify such temporal relationships, conducting a cohort study would be required.” (page 10, lines 291 and 292)

Comment 3: Also, the reliance on self-reported data introduces potential bias.

Response: In response to your suggestion, we have revised the limitations section as follows:

Fourth, the use of self-reported lifestyle questionnaires may introduce some information bias.” (page 10, lines 302 and 303)

Comment 4: The authors should consider using a longitudinal design to establish cause-and-effect relationships.

Response: Thank you for your valuable comment. We believe this concern has been addressed in our response to Comment 2.

Comment 5: Objective dietary assessments should be used to reduce potential bias from self-reported data.

Response: We have acknowledged the potential bias associated with self-reported dietary data while emphasizing the validation of the FFQ. The limitations section has been revised as follows:

Third, although the FFQ has been validated for dietary intake amounts through a 4-day dietary record [34], the use of self-reported dietary questionnaires may still have led to inaccurate estimations of dietary intake.” (page 10, lines 300-302)

Comment 6: Further research should examine the quality of meals and the role of social interactions during meals, as these factors might influence nutrient intake more than companionship alone.

Response: Thank you for your valuable and important feedback. Based on your suggestion, we have revised the manuscript to emphasize the need for future research on dietary quality and the role of social interactions during meals. Additionally, we have incorporated two relevant references to support this point.

While this study focused on the association between the number of dining companions and the amount of dietary intake, dietary quality, including dietary diversity, has also been reported to be inversely related to the risk of cancer and all-cause mortality among older adults [60, 61]. The positive association between the number of dining companions and the amount of fats and oils intake remained even after further adjusting for energy intake in the regression model (data not shown); however, future studies are needed to clarify the association between the number of dining companions and dietary quality. Furthermore, while this study hypothesized social facilitation as a potential mechanism, future research is needed to focus on the role of social interactions during meals.” (page 10, lines 304-312)

References:

  1. Otsuka, R.; Tange, C.; Nishita, Y.; Kato, Y.; Tomida, M.; Imai, T.; Ando, F.; Shimokata, H. Dietary Diversity and All-Cause and Cause-Specific Mortality in Japanese Community-Dwelling Older Adults. Nutrients. 2020, 12.

  1. Tao, L.; Xie, Z.; Huang, T. Dietary diversity and all-cause mortality among Chinese adults aged 65 or older: A communi-ty-based cohort study. Asia Pac J Clin Nutr. 2020, 29, 152-160.